# Screening for Cervical Cancer and Early Treatment (SCCET) Project—The Programmatic Data of Romanian Experience in Primary Screening for High-Risk HPV DNA

**DOI:** 10.3390/diagnostics15162066

**Published:** 2025-08-18

**Authors:** Gabriel Marian Saveliev, Adriana Irina Ciuvică, Dragos Cretoiu, Valentin Nicolae Varlas, Cristian Balalau, Irina Balescu, Nicolae Bacalbasa, Laurentiu Camil Bohiltea, Nicolae Suciu

**Affiliations:** 1Department of Obstetrics and Gynecology, Polizu Clinical Hospital, Alessandrescu-Rusescu National Institute for Mother and Child Health, 020395 Bucharest, Romania; savelievg@yahoo.com (G.M.S.); adrianaciuvica@yahoo.ro (A.I.C.); nicolaesuciu@yahoo.ro (N.S.); 2Department of Obstetrics and Gynecology, Carol Davila University of Medicine and Pharmacy, 020021 Bucharest, Romania; 3Department of Medical Genetics, Carol Davila University of Medicine and Pharmacy, 020021 Bucharest, Romania; dragoscretoiu@yahoo.ro (D.C.); laurentiu.bohiltea@umfcd.ro (L.C.B.); 4Fetal Medicine Excellence Research Center, Alessandrescu-Rusescu National Institute for Mother and Child Health, 020395 Bucharest, Romania; 5Department of Obstetrics and Gynecology, Filantropia Clinical Hospital, 011132 Bucharest, Romania; 6Department of General Surgery, Carol Davila University of Medicine and Pharmacy, 020021 Bucharest, Romania; drbalalau@gmail.com (C.B.); irina_balescu206@yahoo.com (I.B.); nicolaebacalbasa@gmail.com (N.B.); 7Department of General Surgery, “St. Pantelimon” Emergency Clinical Hospital, 020021 Bucharest, Romania; 8Department of Visceral Surgery, Center of Excellence in Translational Medicine, Fundeni Clinical Institute, 022328 Bucharest, Romania; 9Department of Visceral Surgery, Center of Digestive Diseases and Liver Transplantation, Fundeni Clinical Institute, 022328 Bucharest, Romania

**Keywords:** high-risk human papillomavirus, cervical cancer screening, cervical cancer, SCCET, Pap test

## Abstract

**Background/Objectives:** Cervical cancer (CC), caused mainly by high-risk human papillomavirus (hrHPV), remains a global health challenge despite being preventable. The disease’s incidence and mortality rates significantly vary across regions, highlighting the need for effective screening programs. The World Health Organization prioritizes CC screening to monitor and eliminate the disease. The Screening for Cervical Cancer and Early Treatment (SCCET) project aligns with this goal by adhering to the 2012 National Program for Cervical Cancer Screening and implementing the European Guidelines of Quality Assurance. **Methods:** The SCCET initiative facilitates access to equitable and high-quality preventive medical services for Romanian women, incorporating the Babeș–Papanicolaou smear (Pap test) and/or hrHPV DNA screening. Focused on the Muntenia Region of South Romania, the project leverages a methodical approach to gather substantial medical data on hrHPV infection rates and cervical lesions, thereby improving health management for women in the screening program. **Results:** Through public information and educational campaigns about HPV and its link to CC, the SCCET project has significantly enhanced participation in the screening program. In the study conducted between September 2022 and March 2023, 14,385 women aged 30 to 64 years voluntarily participated; of these, 11,996 (83.4%) underwent primary hrHPV DNA screening and were tested using the PowerGene 9600 Plus Real-Time polymerase chain reaction (PCR) system and the commercial Atila BioSystems AmpFire^®^ HPV Screening 16/18/HR test, version 4.1. This substantial participation indicates a positive shift in public attitudes towards CC prevention and highlights the success of the project’s outreach efforts. The study revealed an overall prevalence of hrHPV infection of 12.24%; of these, the most common genotype was other hrHPV types (9.84%), followed by HPV 16 (2.3%) and HPV 18 (0.71%). **Conclusions:** The SCCET project’s recent data on primary hrHPV DNA screening showcases its pivotal role in advancing the management and prevention of CC in Romania. By providing accessible, high-quality screening services and fostering public education on HPV, the initiative has made significant strides toward reducing the burden of CC. This effort aligns with global public health goals, and providing updated information on the prevalence of hrHPV types will allow the development of personalized national screening and vaccination programs to eradicate CC.

## 1. Introduction

Worldwide, human papillomavirus (HPV) infection is currently the most common sexually transmitted viral disease in both men and women. However, the true incidence is not fully known due to the lack of reporting in national registries. Each year, within the European Region of the World Health Organization (WHO), more than 66,000 women receive a new diagnosis of cervical cancer (CC), resulting in the unfortunate demise of over 30,000 women due to this preventable ailment. CC continues to be prominently ranked as one of the most prevalent gynecological malignancies worldwide, currently occupying the fourth position among all cancer types. CC, characterized by its challenging nature and traumatic consequences, predominantly affects women aged 45 to 60, although it can also affect younger individuals. At the regional level, the proportion of invasive CC cases diagnosed in the early stages varies from 35% to 80%, depending on the country, with a five-year survival rate ranging from 54% to 80% [1].

In Romania, according to the study by Bruni et al., it is estimated that 3308 women are diagnosed with CC annually, with a reported number of 1743 fatalities caused by it [2]. The incidence rate of CC, at 33 cases per 100,000 people, is three times higher in Romania than at the European Union level (12 per 100,000), which explains the alarming position of the increased mortality rate [3]. Currently, Romania still occupies the first position in Europe, with a mortality rate of 10.84 per 100,000 [4,5]. Globally, the incidence of CC is 14.1 per 100,000, and the mortality rate is 7.1 per 100,000 [6].

Several factors underlie these high rates, including insufficient implementation of human papillomavirus (HPV) vaccination and screening programs, lack of adequate knowledge of the risk of HPV infection through inadequate public health campaigns, hesitancy to vaccination, economic and socio-cultural barriers, and lack of accurate identification of disadvantaged ethnic communities [2,7,8,9].

The comparative assessment between 2018 and 2022 of the databases on the evolution of cervical cancer incidence and mortality rates, at the level of the four main regions of Europe, did not reveal significant changes in mortality rates. Significant changes were observed in the incidence rates of cervical cancer in the southern and northern European regions, which require continued efforts to prevent CC at the EU level [10,11,12] (Figure 1).

Papillomaviruses, members of the Papovaviridae family, are relatively small viruses, with a diameter of 55 nm, without an envelope, which have an icosahedral capsid structure formed by 72 capsomeres. This contains two proteins (L1 and L2) at the capsid level.

Thus, the HPV genome is formed by a double-stranded circular DNA, functionally formed by three regions. The first represents an upstream non-coding regulatory region, which contains the central promoter and p97 enhancer sequences with a role in regulating DNA replication. The second region is formed by the ORFs E1, E2, E4, E5, E6, and E7, with a role in viral replication and the oncogenesis process. The last region has a role in encoding the L1 and L2 proteins at the capsid level [13].

HPV infection, through the action of the host cell DNA polymerase, promotes replication of its DNA genome, reduces the rate of nucleotide polymorphisms, and promotes oncogenic transformation by inhibiting the activity of the p53 tumor suppressor protein and the apoptosis process by the E6 oncoprotein [14]. Currently, genomic analysis of HPV with cutaneous and mucosal tropism (preferably cervical) has described over 400 genotypes (International Reference Center for Human Papillomavirus) with high (hr) and low (lr) oncogenic risk. Of these, over 40 genotypes (hrHPV ± lrHPV) can infect the genital mucosa and skin, most commonly causing benign, borderline, and malignant cervical lesions [15]. hrHPVs are responsible for the development of cervical and anal cancers, most vulvar, vaginal, and penile cancers, as well as some types of head and neck cancers (oral cavity, oro/hypopharyngeal, and laryngeal) [14].

The relative contributions of HPV 16/18 (highly oncogenic) and HPV 6/11/16/18/31/33/45/52/58 genotypes of over 90%, single or in association, are proven in the pathogenesis of CC [16]. The Babeș–Papanicolaou smear (Pap test), which has been applied since 1941, and implicitly, the Pap cervical screening, represented the first step in the early detection of CC. However, the demonstration (in the 1980s) that HPV is the primary etiology of CC led to the introduction of the HPV vaccine and HPV testing as the first intention of cervical screening. High-risk HPV identification and liquid cytology are common tests recommended and widely used worldwide. Certification of hrHPV etiology accelerated the structuring and implementation of a comprehensive global strategy for preventing, controlling, and eliminating CC [17].

More than 99% of CC cases result from persistent infections with high-risk genotypes, notably HPV 16 and HPV 18, commonly transmitted through unprotected and random sexual encounters [18,19,20]. The identification of oncogenic hrHPV strains prompted the development of prophylactic vaccines that are both safe and effective. To date, six vaccines have received authorization for clinical use and commercialization. Typically, three types of vaccines offer protection against specific hrHPV strains, complementing existing screening programs capable of early disease detection, thereby facilitating efficient CC treatment and management—a goal achieved in highly developed economies [21,22,23].

The use of HPV/hrHPV DNA testing in the identification and management of CC is supported by several lines of evidence:(a)Persistent hrHPV infection is implicated in the development of CC.(b)The risk of developing CIN2+ in the long term in women with a negative hrHPV test (sensitivity ~95%) is much lower compared to those with negative cytology.(c)Testing can be performed using various cervical biological specimens.(d)Testing is objective and has greater global consistency compared to cytology.(e)hrHPV genotypes, especially HPV 16/HPV 18, are highly oncogenic [22,24,25,26,27].

Currently, cervical cancer screening involves either primary cytology, primary HPV testing, or co-testing (cytology and HPV testing) [28]. Although HPV testing has been shown to have real benefits, with many health systems recommending this type of primary screening instead of cytology, there are also several limitations, especially in developing countries. Cervical cancer screening, according to the study by Bruni et al., was identified in 139 of 202 countries (69%); of these, only 48 (35%) recommend primary HPV-based screening [29]. The goals of CC screening and treatment are to reduce the disease incidence and mortality rates by identifying women with precancerous lesions [1,7,30]. Moreover, screening for CC constitutes a priority objective for the WHO and is an essential indicator for disease eradication.

The medical strategy to combat CC primarily centers on primary and secondary prevention, with primary prevention and screening representing the most effective methods for reducing mortality rates. Disparities exist in screening rates, early diagnosis, and treatment of CC, particularly in regions with lower economic resources, ethnic variations, and age disparities [15,31,32,33].

Although in 2012, the Romanian Ministry of Health introduced a national screening program targeting women aged 25 to 64, the lack of a national registry for monitoring CC and a unified health policy has failed to significantly reduce the mortality rate caused by CC [34]. The primary screening methods include cytology and HPV testing every 5 years. As of 2013, the program achieved an invitation coverage of 65%, an examination coverage of 9.2%, and a participation rate of 14.2% [35]. In 2018, the proportion of women screened in Romania was approximately 25% [36].

The Screening for Cervical Cancer and Early Treatment (SCCET) project contributes to the 2012 National Program for Uterine Cervical Cancer Screening, employing the European Guides of Quality Assurance. This initiative, which was carried out from August 2020 to November 2023, aimed to provide accessible, equitable, high-quality preventive healthcare, including Pap smear and/or hrHPV testing for Romanian women. By applying the SCCET methodology in pre-defined geographic regions, particularly the Muntenia Region in south Romania, substantial medical data concerning hrHPV infection rates and cervical lesions have been generated. These data have positively influenced the healthcare management of women participating in the screening project [2].

This paper presents recent data from the SCCET project spanning 2022–2023, focusing specifically on the primary screening of DNA HPV/hrHPV, which has significantly impacted CC prevention management. Participants from seven counties in the studied region were tested for HPV, and those with positive results for hrHPV were investigated cytologically. Finally, hrHPV+ patients with positive cytology were referred for colposcopic examination and appropriate therapeutic management according to indications. The importance of the study is highlighted by the provision of valuable epidemiological information in an incompletely studied geographical region, which will help health services correctly manage resources and make decisions to increase the number of vaccinated people.

## 2. Materials and Methods

### 2.1. Study Design and Participant Characteristics

We developed a prospective clinical study to evaluate the efficacy of the primary screening of DNA HPV/hrHPV for the age group of 30–64 years, according to the methodology of the current medical guidelines on screening for CC. This study was carried out in seven counties (Prahova, Argeș, Dâmbovița, Teleorman, Călărași, Giurgiu, Ialomița) from the Muntenia Region of south Romania affiliated with the SCCET, in the timeframe of September 2022 to March 2023 (7 months). Of the 14,385 participating women tested primarily for hrHPV, 11,996 (83.05%) were included in the final analysis, the rest being excluded due to lack of additional information or pregnancy (Figure 2).

Each woman involved in the SCCET project received prior information regarding cervical hrHPV infections and their role in the initiation and progression of CC. The exclusion criteria were prior treatment with excision/ablation methods for cervical precancerous lesions, age under 30 or over 65, and history of hysterectomy or immunosuppression. The sampling method was consecutive.

### 2.2. Ethical Approval

The Research Medical Ethics Committee of the “Alessandrescu-Rusescu” National Institute for Mother and Child Health approved the survey protocol (No. 7315/31.03.2023). The ethical standards of the Declaration of Helsinki were followed. All patients signed the informed consent form before HPV DNA testing.

### 2.3. Clinical Evaluation and Data Collection

Participants in the screening program completed structured SCCET questionnaires, which included personal and medical information accompanied by confidentiality clauses and informed consent for voluntary participation. The questionnaire covered various personal details such as age, urban or rural residence, hormonal status (cyclic or menopause), the presence of intrauterine devices (IUDs), and a history of sexually transmitted infections (STIs).

Biospecimens, comprising cervical swabs/lavages, were collected with informed consent following SCCET protocols. Thus, the containers with the collected samples allow storage at room temperature for 30 days. These specimens were transported in optimal conditions (with insulated refrigerated boxes at +4 °C), following biological sample transportation rules and SCCET protocols, to the “Alessandrescu-Rusescu” National Institute for Mother and Child Health from Bucharest, the leading partner in SCCET. The samples collected from mobile units or public hospitals were processed and analyzed in the institute’s genetics laboratories.

Thawed samples were processed by centrifugation for 30 min to remove the supernatant, and the contents were transferred to a tube and subsequently incubated using the PowerGene 9600 Plus real-time polymerase chain reaction (PCR) system (Atila Biosystems, Inc., Sunnydale, CA, USA) [37]. Finally, genotyping was performed using Atila BioSystems AmpFire^®^ HPV Screening 16/18/HR commercial test (Atila Biosystems, Inc., Sunnydale, CA, USA), version 4.1 (an isothermal multiplex amplification test) by fluorescent detection, which allows the detection of 15 high-risk HPV genotypes [38].

The Babeș–Papanicolaou test was performed on the participants who tested positive after primary hrHPV testing. After collection and transport, cytology smears were processed, stained, and validated according to the Bethesda system by local accredited laboratories.

### 2.4. Statistics

Data analysis was performed using the SPSS^®^ 27.0 software (SPSS Inc., Chicago, IL, USA). We described the continuous variables using the median (range), mean, and standard deviation with a 95% CI or count (percent) when appropriate. The chi-square test was performed to investigate the linear trend in hrHPV prevalence and age groups, as well as hrHPV positivity. We evaluated the subgroups using Student’s *t*-test for the variables not normally distributed and Fisher’s exact test. Medical geneticists interpreted and validated the results, which were considered statistically significant with *p* < 0.05.

## 3. Results

### 3.1. Demographic Distribution of the Study Group

In the present study, we examined the prevalence of hrHPV among 11,996 women aged 30–64. The participants were categorized based on age, residence, hormonal status, presence of intrauterine devices (IUDs), and history of sexually transmitted infections (STIs). The analysis of the survey questionnaires generated the study group data, which is detailed in Table 1.

The demographic distribution of the study cohort was diverse. Most cases (6522 or 54.36%) were from rural areas, while urban cases accounted for 5474 (45.64%). From a hormonal point of view, the rate of participants in the active phase of the menstrual cycle (90.28%) was significantly higher compared to those in menopause (8.39%).

The analysis of hrHPV testing revealed that 1468 women (12.24%) tested positive for hrHPV. Correlated with various risk factors, dependent on cervical hrHPV infection, the distribution of hrHPV+ cases is detailed in Table 2.

### 3.2. Risk Factors Correlated with hrHPV Infection

When examining hrHPV+ cases concerning the risk factors, it was found that the rural population had a slightly higher prevalence rate of 53.62% compared to 46.38% in urban areas. Regarding hormonal status, 91.55% of the women had a cyclic hormonal status, 5.92% were in menopause, and the differences reported in the hrHPV presence rate are insignificant (*p* = 0.508). The presence of IUDs was noted in 45 (3.06%) cases, and a history of STIs was documented in 86 (5.85%) cases. Additionally, the coexistence of IUDs and history of STIs was seen in 1.84% of hrHPV+ cases.

### 3.3. HPV Testing and Genotyping

In this cohort of study participants, the prevalence of hrHPV+ cases tested from cervical samples was 12.24%. HPV distribution (single/multiple infections) following genotyping identified HPV 16 in 276 (2.3%) cases, HPV 18 in 86 (0.71%) cases, and other hrHPV strains in 1181 (9.84%) cases. In total, in 95.23% (1398/1468) of the cases, a single hrHPV type was identified in the cervical swab, two types in 4.63% (68/1468) of cases, and multiple infections in 0.13% (2/1468) of cases (Figure 3).

### 3.4. Prevalence of hrHPV Infection

The geographical distribution of the prevalence of hrHPV+ cases in the counties of the Muntenia Region of South Romania revealed increased values in Călărași (26.85%) and Giurgiu (14.35%) counties (Figure 4 and Figure 5).

Within SCCET, educational campaigns on the impact of cervical hrHPV infections in the initiation and progression of CC have been effective in encouraging participation and hrHPV ± Pap testing among 11,996 women. Notably, women aged 30 to 64 years, following the SCCET protocols, underwent hrHPV testing as the preferred primary screening method for CC, with an hrHPV positivity rate of 12.24%. When co-testing Papanicolaou of hrHPV+ cases, 131 (8.92%) had cytological changes as follows: a total of 60 (45.8%) patients had low-grade cytology (L-SIL/ASC-US), and 71 (54.2%) patients had high-grade cytology (ASC-H, AGC, H-SIL) (*p* = 0.592). The increased rates regarding the association of high prevalence of HPV infection and abnormal cytology in hrHPV+ patients may be due to the lack of the correct CC screening and the relatively late implementation of the screening and vaccination programs.

Also, the mean age of the women participating in the study was 42.38 ± 3.21 years, being lower in women positive for hrHPV (42.18 ± 3.27 years) compared to those with a negative test for hrHPV (42.41 ± 3.2 years) (*p* = 0.107) (Figure 6).

### 3.5. Implications for Public Health

The results suggest that hrHPV infection is fairly prevalent among the study population, with a significant distribution across different demographic and physiological risk factors. The findings underscore the importance of targeted screening and vaccination programs, especially in areas and populations with higher prevalence rates.

## 4. Discussion

The programmatic data revealed within this project provides important data on the prevalence of hrHPV in the Muntenia South Region of Romania among the population aged between 30 and 64 years. The study will ultimately be reported at the national level to identify geographical variations regarding the prevalence of hrHPV.

Primary HPV screening is not recommended for women under 30 years of age due to low specificity [39], and a negative result after the age of 55 is associated with a reduced risk of CC [40]. In the 30–64 age group, the global incidence rate of CC is 29.6 per 100,000 and the mortality rate is 13.4 per 100,000, making it the second leading cause of death after breast cancer. At the European level, the incidence rate is 21.9 per 100,000 and the mortality rate is 7.6 per 100,000, making it the fourth leading cause of death after breast, lung, and colorectal cancer. In Romania, the incidence rate of CC is 45.4 per 100,000, and the mortality rate is 17.4 per 100,000, making it the second leading cause of death after breast cancer [6].

The increased prevalence of hrHPV is correlated with several risk factors such as HPV genotype, geographic distribution, ethnicity, age, comorbidities, low socio-economic status, and cultural differences [15]. Thus, a very high variability in hrHPV prevalence was observed at the European level, from 21.4% in eastern Europe to 9.0% in western Europe. Compared to the countries of eastern Europe, the situation is as follows: Hungary (11.1%), Poland (14.4%), Lithuania (24.2%), and the Czech Republic (25.6%) [30]. Our study highlighted that the data on the overall prevalence of hrHPV infection (12.24%), as well as those reported for HPV 16 (2.3%), HPV 18 (0.71%), and other hrHPV types (9.84%), are similar to those published by the study of Song et al., which shows a prevalence of 11.1% and 1.6%, 0.6%, and 8.9%, respectively, depending on the HPV type [41]. Stratification of hrHPV+ cases was 11.65% single HPV infections and 0.58% multiple HPV infections, results similar to those obtained by Gao et al.—10.55% and 1.75%, respectively [42].

An increased prevalence of infection with other hrHPV genotypes (9.84%) was observed in the studied cohort, possibly due to the decrease in the rate of HPV16/18 infections secondary to the protection provided by vaccination. This result is similar to that of other studies [43,44]. Although HPV16/18 strains still pose a high risk, the increased prevalence of other hrHPV types will require reconsideration of the cervical screening strategy. The increased prevalence of other hrHPV types requires additional research to identify each type of hrHPV to monitor the evolution and to reconsider the vaccination strategy by introducing other strains of hrHPV. Thus, the geographical distribution of the prevalence of hrHPV infection, with the analysis of HPV types, allows the harmonization of HPV screening programs with vaccination programs.

The age-stratified prevalence of hrHPV+ compared to the general population reached its maximum in the age group <40 years (13.65%). At the European level, epidemiological studies on hrHPV prevalence rates in general populations have found high percentages in women aged <35 years and lower rates in women aged >45 years, with an increase in those aged >60 years. The increase in the age group <35 years may be due to changes in sexual behavior in women; in the case of those aged >60 years, the increase may be due to a deficient immune status that can reactivate latent hrHPV infection [45]. Another study revealed that the highest positive detection rate of hrHPV infection was observed in women aged ≥50 years (due to a low immune response) [41].

Although, apparently, the prevalence of hrHPV is not that high compared to other countries, mortality from CC is very high, which represents low population participation in CC screening, a lack of adequate monitoring, and an ineffective national strategy. The causes of increased mortality are represented by a low vaccination rate, the degree of implementation of screening programs, and therapeutic strategies after screening.

The data presented earlier highlight that urban residence and an active menstrual cycle status were significant risk factors for cervical hrHPV+ infection. The vulnerability of rural women can be attributed to environmental factors, societal norms, traditionalism, and varying degrees of health education and sexual education.

Additionally, women with active menstrual cycles tend to engage in more sexual activity, consequently increasing the risk of acquiring hrHPV. Multiple sexual partners, random sexual encounters, and a history of STIs further elevate the risk of cervical infection with hrHPV. Conversely, studies have indicated that women with IUDs implanted have a lower risk of cervical hrHPV infection [2]. Giubbi et al. reported that sexually transmitted infections (STIs) are more frequently associated with hrHPV+ women, without establishing a possible role in the development of high-grade cervical lesions. The mechanisms by which STIs may cause the formation of lesions in the cervical epithelium favor the intracellular persistence of HPV through a proinflammatory status and initiate an ulcerative process by decreasing the immune response [46].

Due to the association of STIs with hrHPV+ patients and the increased incidence of STIs, the opportunity to perform molecular screening for STIs in patients included in CC screening programs can be evaluated. Thus, the low rate of hrHPV positivity observed in patients > 40 years is also associated with a much lower risk of having an STI secondary to sexual education and a lower number of sexual contacts.

Women with positive cervical hrHPV+ tests, as per SCCET protocols, are informed that the results indicate a high risk of initiating precursor lesions of CC, necessitating further clarifying investigations following international protocols and guidelines, with Pap testing and colposcopy being the initial steps [2]. Thus, in patients with hrHPV+, the association of positive cytology was 8.92%.

The study by Ilisiu et al., conducted between July 2015 and April 2017 on a group of 2060 women aged between 18 and 70 years, established a prevalence of hrHPV+ of 16.9% at the level of 24 counties in four regions (north, center, west, and south) of Romania. The highest prevalence rate of hrHPV was observed in the west (23%) and north (19.2%) regions. Regarding the ethnic distribution, the highest prevalence rate was identified in Romanians (17.9%) and the lowest for the Roma population (7.8%). Also, an increased prevalence was observed in urban areas compared to rural areas, as in our study [34]. The prevalence in the south region, which was 13.1%, is close to that obtained in our study.

Currently, hrHPV tests used for CC screening and management are predominantly high-throughput molecular tests capable of detecting 13–14 high-risk types. Although targeted hrHPV identification can provide risk insights, practical considerations often lead screening programs to manage women based on positive/negative hrHPV testing, especially in programs aligning with medical guidelines [1,7,31,35]. Over the past decade, there has been a considerable proliferation of commercially available HPV tests; however, the quantity and quality of evidence regarding their clinical performance vary considerably. A series of studies have shown that, for samples collected directly, the AmpFire^®^ HPV Screening 16/18/HR test generated results and accuracy similar to Cobas and SeqHPV regarding histological CIN2+, a fact that justifies its performance and, implicitly, the choice for testing hrHPV in primary screening [38,47]. The choice of an assay that can be integrated into women’s healthcare systems may be influenced by pragmatic attributes such as cost, ease of use, throughput, and the adaptability of the medical platform for other applications [18,48].

The three primary accepted indications for hrHPV testing are as follows:(a)Risk stratification/triage for equivocal or low-grade cytological abnormalities, directing individuals to colposcopy;(b)Monitoring therapeutic success following cervical lesion removal, often referred to as a “test of cure”;(c)Primary screening for CC.

The first two indications have been established in opportunistic CC screening programs over several years.

Primary screening for CC through hrHPV testing, a more recent paradigm shift, is now recognized as the optimal screening modality, with more countries embracing this approach [22,33,49,50]. Furthermore, in a retrospective analysis of 924 postmenopausal women, Kiff et al. demonstrated that hrHPV testing alone may have greater accuracy than co-testing [51].

Since in many countries, national CC screening programs encounter a series of difficulties, attempts are being made to switch from screening based on cervical cytology to one based on hrHPV genotyping, with a risk stratification related to age and associated risk factors. Ultimately, targeting health system resources would allow for identifying risk groups, providing correct prevention, and developing personalized screening. Although the prevalence rate of hrHPV+ in the 30–64 age group, where primary HPV testing is performed, is not as high as in other countries, Romania is facing an increased incidence and mortality from CC. This situation represents a public health problem and requires better mobilization of patients in screening programs through better information for them.

The strategic objectives of the SCCET project aim to improve education and provide optimized prevention, diagnosis, monitoring, and treatment services:(a)Enhancing women’s health through the implementation of a comprehensive screening program in the seven counties within the Muntenia Region of South Romania (Prahova, Argeș, Dâmbovița, Teleorman, Călărași, Giurgiu, Ialomița). This involves identifying susceptible women among the participants through consultations with Obstetrics-Gynecology specialists, coupled with diagnostic, treatment, counseling, monitoring, and prevention services;(b)Establishing a robust personal data protection system in compliance with prevailing legislation, with stringent measures for secure data processing throughout the project;(c)Conducting information and education sessions to raise awareness and promote the necessity of participating in the screening program. This is targeted at specific groups and the general public and emphasizes the correlation between access to medical services, awareness of health status, and women’s rights to healthcare.(d)Expanding women’s access to high-quality medical services [2].

However, the routine use of HPV/hrHPV testing for CC screening also presents potential limitations and disadvantages. Given the sensitivity of hrHPV testing, concerns arise that it may lead to unnecessary cervical surgery for transient HPV infections and less severe cervical lesions that might resolve spontaneously, potentially subjecting women to unnecessary physical and psychological trauma. Additionally, the socio-economic conditions in some countries may not support the routine use of HPV testing in screening programs.

The use of self-report questionnaires is a bias factor, which can be mitigated by collecting data from many study participants. Another limitation of this study is the fact that the data presented are partial; they are part of a 3-year structured project that did not analyze the stratification of cases by HPV types but only carrying out the formal identification of HPV with high risk at the population level related to the seven counties studied and correlation with the positive Pap test.

A strength of the study is represented by the relatively large sample size, which demonstrates adequate statistical power regarding the prevalence of hrHPV in the general population by age-stratified groups. Another key point is represented by the epidemiological value, which provides important information for an incompletely studied geographical region, complementing the national-level data for better resource management.

Guidelines for CC screening programs emphasize the importance of patient and healthcare practitioner education as a crucial component of implementing primary hrHPV testing [33,52,53].

## 5. Conclusions

Screening programs for CC that integrate high-quality diagnostic and treatment services have significantly reduced mortality rates, as demonstrated in randomized controlled trials conducted in different countries. To date, the SCCET has achieved a high participation rate among women aged 30–64 years, with an average hrHPV+ prevalence of 12.24% across the geographical region studied, supporting the recommendation of HPV/hrHPV testing as the primary screening method for CC. These findings highlight the need for patient-directed communication strategies to increase the routine implementation of primary HPV/hrHPV testing for CC screening, ensuring awareness of its effectiveness and evidence-based benefits. Furthermore, integrating informed screening preferences into patient–physician discussions can promote patient-centered care and evidence-based medical practices. Thus, highlighting differences in hrHPV prevalence between the counties studied in the project and identifying those with increased prevalence and population groups at highest risk by age allows for better coordination of resources with an impact on public health. Providing updated information on the prevalence of hrHPV types will allow the development of personalized national screening and vaccination programs to eradicate CC.

## Figures and Tables

**Figure 1 diagnostics-15-02066-f001:**
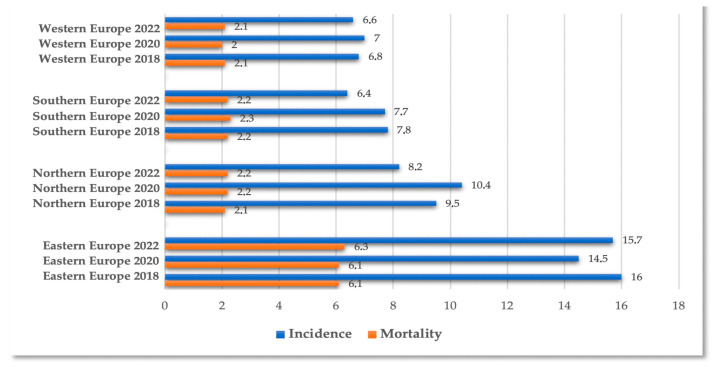
Evolution of incidence and mortality rates of cervical cancer in 2018, 2020, and 2022 in four regions of Europe—data provided by Globocan source [10,11,12].

**Figure 2 diagnostics-15-02066-f002:**
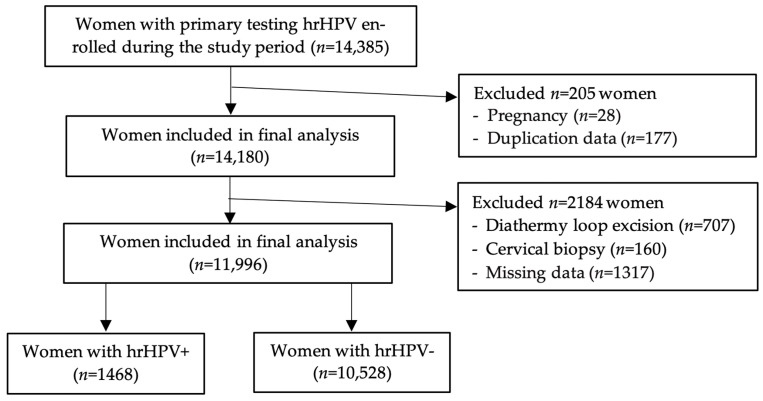
Flow diagram of patient distribution.

**Figure 3 diagnostics-15-02066-f003:**
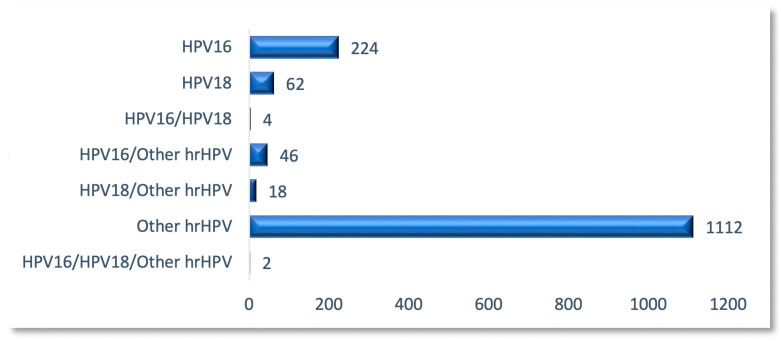
HPV genotyping distribution in the study group.

**Figure 4 diagnostics-15-02066-f004:**
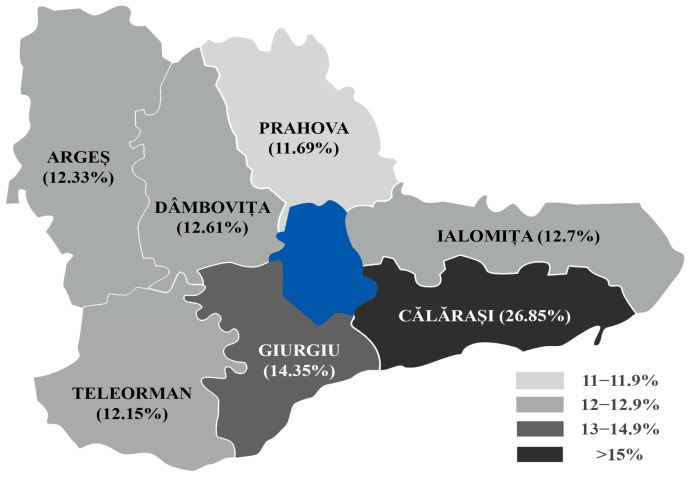
Geographical distribution of the prevalence of hrHPV-positive cases in each studied county (note—Bucharest, the capital of Romania, is marked in blue and is not included in this study).

**Figure 5 diagnostics-15-02066-f005:**
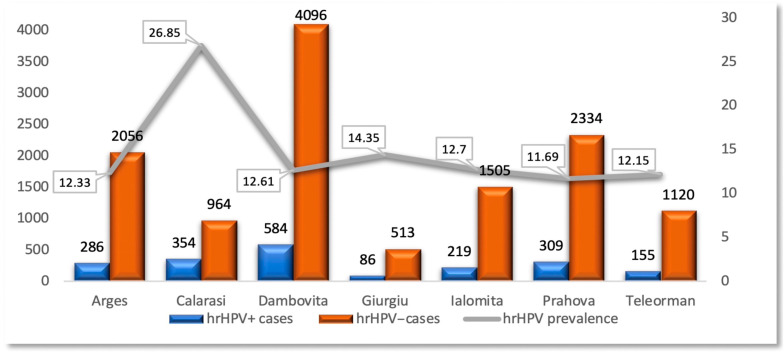
Distribution of the hrHPV-/hrHPV+ cases and hrHPV prevalence in each studied county.

**Figure 6 diagnostics-15-02066-f006:**
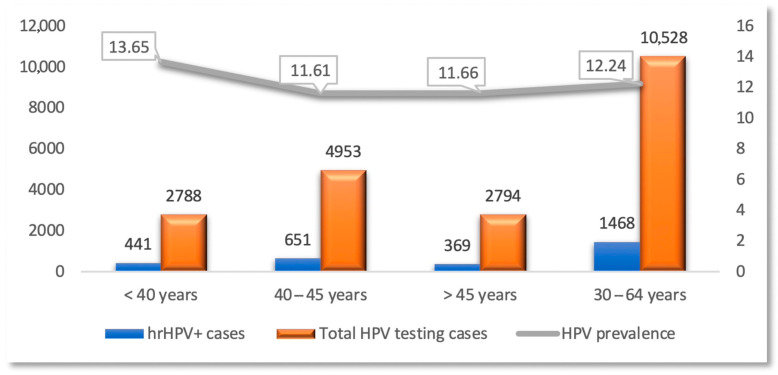
Distribution of the hrHPV−/hrHPV+ cases and HPV prevalence according to age stratification.

**Table 1 diagnostics-15-02066-t001:** Clinical characteristics of the study group.

Parameter	Total hrHPV Testing Cases
Number of cases (*n*)	11,996
Age (mean ± SD) years	42.39 ± 3.21
Residence (*n*, %) - urban - rural	5474 (45.64%) 6522 (54.36%)
Hormonal status (*n*, %) - active menstrual cycle - menopause - other	10,831 (90.28%) 1007 (8.39%) 158 (1.32%)
IUDs (*n*, %)	390 (3.25%)
STIs (*n*, %)	592 (4.93%)

**Table 2 diagnostics-15-02066-t002:** hrHPV infection patterns correlated with various risk factors.

Parameter	hrHPV Test		hrHPV Infection Types			
	hrHPV−	hrHPV+	HPV 16	HPV 18	Other hrHPV	*p*-Value
Number of cases (*n*, %)	10,528 (87.76%)	1468 (12.24%)	276 (2.3 %)	86 (0.71%)	1181 (9.84%)	-
Age (mean ± SD)	42.41 ± 3.2	42.18 ± 3.27	41.77 ± 3.3	42.74 ± 3.38	42.17 ± 3.25	0.107 *
Residence (*n*, %) - urban - rural	4773 (50.11%) 5755 (49.89%)	681 (46.38%) 787 (53.62%)	127 (46.01%) 149 (53.99%)	44 (51.16%) 42 (48.84%)	550 (46.57%) 631 (53.43%)	0.534 **
Hormonal status (*n*, %) - active menstrual cycle - menopause - other	9487 (90.12%) 901 (8.55%) 140 (1.33%)	1344 (91.55%) 87 (5.93%) 37 (2.52%)	260 (94.2%) 11 (3.99%) 5 (1.81%)	77 (89.53%) 8 (9.3%) 1 (1.17%)	1076 (91.11%) 89 (7.54%) 16 (1.35%)	<0.508 **
IUDs (*n*, %)	345 (2.87%)	45 (3.06%)	8 (2.89%)	3 (3.48%)	34 (2.87%)	N.S.
STIs (*n*, %)	506 (4.21%)	86 (5.85%)	19 (6.88%)	4 (4.65%)	63 (5.33%)	N.S.

Note: IUD—intrauterine device; STIs—sexually transmitted infections; SD—standard deviation; N.S.—not significant; * Student’s *t*-test; ** Fisher’s exact test.

## Data Availability

Restrictions apply to the availability of these data. Requests to access the datasets should be directed to sccut@insmc.ro.

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
