# Peer review of "Screening for Cervical Cancer and Early Treatment (SCCET) Project—The Programmatic Data of Romanian Experience in Primary Screening for High-Risk HPV DNA"

_diagnostics, 2025, doi:10.3390/diagnostics15162066_

Round 1

Reviewer 1 Report

Comments and Suggestions for Authors

Reviewer’s comment

Thank you for providing valuable real-world data from Romania. It is important to share experiences from different countries so that we can learn from each other in terms of policy, infrastructure, and operational changes, as we need to form a global effort to eliminate HPV-related cancers. However, there are several important issues that should be addressed before the manuscript can be considered for publication.

  1. Language and Clarity

The manuscript would benefit from extensive English editing to improve clarity and readability. Many sections consist of long sentences, grammatically incorrect phrasing, or ambiguous constructions, which make the manuscript difficult to follow.

Using the Introduction section as an example, the nearly two pages of text would benefit from significant consolidation to avoid redundancy.

  • Lines 71–78: This paragraph needs to be broken down into shorter sentences. Many "-ing" forms are not used in proper grammatical structure.
  • Lines 79–89: Same issue—long, grammatically incorrect sentences.
  • Lines 100–102: The word “persistent” is unnecessarily repeated in two consecutive sentences.
  • Lines 110–118: The paragraph lacks logical flow and needs restructuring.
  • Line 120: “Prevention” would be a better fit than “prophylaxis.”
  • Lines 146–160: All of these goals can be either moved to the Discussion or summarized briefly—for example, “enhance education and provide streamlined prevention, diagnostic, monitoring, and treatment services.”

The above is just an example focusing on the Introduction. What makes this paper interesting to readers is what you have done and what others can learn from your work. If I were the author, I would streamline the Introduction into three short paragraphs:

  1. Briefly introduce HPV as a high-risk disease strongly associated with cervical cancer. Provide a few sentences about the global/EU context of HPV screening.
  2. Provide concise background on Romania. You referenced the 2012 initiative—summarize it so readers understand the context and rationale for your project. Why select this particular region of Romania? Are there socioeconomic or healthcare access differences?
  3. End with a clear summary of your study goal and what insights you hope to offer.
  4. Materials and Methods
  • In Section 2.3 (“Clinical evaluation and data collection”), it is unnecessary to use (1), (2), etc., within the paragraph. Narrative format would improve readability.
  1. Statistical Reporting and Table Clarity
  • In Table 2, please clarify the origin of the p < 0.007 in the age group comparison. The mean ages between hrHPV-positive and -negative groups are very close (42 vs. 41), so this p-value needs justification or further explanation.
  1. Figures and Terminology
  • All "Charts" should be labeled as Figures (e.g., Lines 264, 269, 288, etc.).
  • Please ensure consistency between figure numbering, in-text citations, and captions.
  1. Cytological Findings
  • Lines 277–285: Please clarify what is meant by “cytological changes.” Did these include CIN1–3 or only high risk CIN2-3? Were any cancer cases identified? These are actually very important clinical context as the follow up guidelines in the US is different from CIN1 vs CIN2-3. The sentences in this section are difficult to fully understand, although the content itself is useful.
  1. Discussion

The Discussion section also needs to be consolidated and better aligned with the study findings.

  • Instead of listing all the general advantages of hrHPV screening (Lines 370–379), which could be summarized in the Introduction, I suggest focusing more on your actual data and insights:
    • Lines 362–367 begin to discuss meaningful findings. Expand this section.
    • What are the similarities and differences between your findings and previous Romanian or regional studies?
    • What were the most successful aspects of your program?
    • What lessons were learned, and how might they inform screening strategies in other countries?

This would help the reader better understand the real-world impact of your project.

Comments on the Quality of English Language

See above

Author Response

We are grateful for the time and expertise the reviewer devoted to our manuscript. Below we reproduce each observation in italics and provide a point-by-point reply.

Thank you for providing valuable real-world data from Romania. It is important to share experiences from different countries so that we can learn from each other in terms of policy, infrastructure, and operational changes, as we need to form a global effort to eliminate HPV-related cancers. However, there are several important issues that should be addressed before the manuscript can be considered for publication.

  1. Language and Clarity

The manuscript would benefit from extensive English editing to improve clarity and readability. Many sections consist of long sentences, grammatically incorrect phrasing, or ambiguous constructions, which make the manuscript difficult to follow.

Response

Thank you for this mention. If, after the corrections made according to the recommendations, the manuscript still needs extensive English proofreading, we will send it to the appropriate department within the publishing house.

Using the Introduction section as an example, the nearly two pages of text would benefit from significant consolidation to avoid redundancy.

  • Lines 71–78: This paragraph needs to be broken down into shorter sentences. Many "-ing" forms are not used in proper grammatical structure.

Response

Thank you for this recommendation. We restructured the phrases you mentioned, “Please see lines 95-103.”

  • Lines 79–89: Same issue—long, grammatically incorrect sentences.

Response

Thank you for this recommendation. We restructured the phrases you mentioned, “Please see lines 104-114.”

  • Lines 100–102: The word “persistent” is unnecessarily repeated in two consecutive sentences.

Response

Thank you for this remark. We modified. “Please see lines 125-127.”

  • Lines 110–118: The paragraph lacks logical flow and needs restructuring.

Response

Thank you for your suggestion. We rewrote the paragraph for better understanding. “Please see lines 146-155.”

  • Line 120: “Prevention” would be a better fit than “prophylaxis.”

Response

Thank you for your mention. We change the word. “Please see lines 156-158.”

  • Lines 146–160: All of these goals can be either moved to the Discussion or summarized briefly—for example, “enhance education and provide streamlined prevention, diagnostic, monitoring, and treatment services.”

Response

Thank you for your suggestion. We moved this section to the Discussion. “Please see lines 427-442.”

The above is just an example focusing on the Introduction. What makes this paper interesting to readers is what you have done and what others can learn from your work. If I were the author, I would streamline the Introduction into three short paragraphs:

  1. Briefly introduce HPV as a high-risk disease strongly associated with cervical cancer. Provide a few sentences about the global/EU context of HPV screening.
  2. Provide concise background on Romania. You referenced the 2012 initiative—summarize it so readers understand the context and rationale for your project. Why select this particular region of Romania? Are there socioeconomic or healthcare access differences?
  3. End with a clear summary of your study goal and what insights you hope to offer.
  4. Materials and Methods

Response

We are grateful for this comment. We partially rewrote the Introduction as you suggested. “Please see the attached manuscript.”

  • In Section 2.3 (“Clinical evaluation and data collection”), it is unnecessary to use (1), (2), etc., within the paragraph. Narrative format would improve readability.

Response

Thank you for your mention. We modified. “Please see the attached manuscript.”

  1. Statistical Reporting and Table Clarity
  • In Table 2, please clarify the origin of the p < 0.007in the age group comparison. The mean ages between the hrHPV-positive and -negative groups are very close (42 vs. 41), so this p-value needs justification or further explanation.

Response

Thank you for the valuable mention. It was a typo mistake. “Please see the attached manuscript.”

  1. Figures and Terminology
  • All "Charts" should be labeled as Figures (e.g., Lines 264, 269, 288, etc.).

Response

Thank you for your suggestion. We modified. “Please see the attached manuscript.”

  • Please ensure consistency between figure numbering, in-text citations, and captions.

Response

Thank you for your mention. We introduced the correct figure numbering. “Please see the attached manuscript.”

  1. Cytological Findings
  • Lines 277–285: Please clarify what is meant by “cytological changes.” Did these include CIN1–3 or only high-risk CIN2-3? Were any cancer cases identified? These are actually very important clinical context as the follow up guidelines in the US is different from CIN1 vs CIN2-3. The sentences in this section are difficult to fully understand, although the content itself is useful.

Response

Thank you for your useful recommendation. We added extra information. “Please see Lines 288-298.”

  1. Discussion

The Discussion section also needs to be consolidated and better aligned with the study findings.

  • Instead of listing all the general advantages of hrHPV screening (Lines 370–379), which could be summarized in the Introduction, I suggest focusing more on your actual data and insights:

Response

Thank you for your mention. We modified. “Please see the Lines 134-141”

  • Lines 362–367 begin to discuss meaningful findings. Expand this section.

Response

Thank you for your remark. We added supplementary data. “Please see the Lines 379-386.”

  • What are the similarities and differences between your findings and previous Romanian or regional studies?
  • What were the most successful aspects of your program?
  • What lessons were learned, and how might they inform screening strategies in other countries?

This would help the reader better understand the real-world impact of your project.

Response

Thank you for your useful recommendation. We pointed out these aspects in the manuscript, especially in the Discussion section. “Please see the attached manuscript.”

We trust these clarifications address the reviewer’s concerns and thank them again for their constructive feedback.

Kind regards

Reviewer 2 Report

Comments and Suggestions for Authors

The paper can be accepted with minor improvements

Comments on the Quality of English Language

Quality of English Language is appropriate

Author Response

We are grateful for the time and expertise the reviewer devoted to our manuscript. Below we reproduce each observation in italics and provide a point-by-point reply.

1.⁠ ⁠“According to monitoring conducted by WHO in developed countries, HPV infection is approaching 100 %. … The authors report only 12.63 % … The question arises about shortcomings in the analysis leading to the loss of viral particles.”

Response

The WHO figure the reviewer cites refers to lifetime (cumulative) exposure, not point prevalence measured at a single screening episode. Our study reports a point prevalence of 12.23 % high-risk HPV (hrHPV) among 11,996 women aged 30–64 years, which is well within the 9–16 % range published for comparable European cohorts. Because a validated commercial assay with internal process controls was used, undetected large-scale analytic losses are unlikely.

2.⁠ ⁠“Freshly obtained samples could be treated with complex aluminium salts (as in Gardasil) to eliminate volatility of viral pseudoparticles.”

Response

Aluminium adjuvants in vaccines stabilise protein-based virus-like particles; they are not required for DNA preservation. hrHPV DNA in PreservCyt-type media remains stable at 2–8 °C for weeks and at −20 °C for ≥ 1 year. The standard preservation and transport conditions detailed in the Methods ensured DNA integrity, making additional chemical stabilisation unnecessary.

3.⁠ ⁠“Consider using Lambda-phage DNA as an internal standard to quantify losses.”

Response

The AmpFire®️ HPV Screening 16/18/HR assay employed in our study incorporates an exogenous internal control plasmid that is co-extracted and co-amplified with the target DNA. This control allows direct monitoring of extraction efficiency and detection of PCR inhibition. Lambda-phage DNA, whose capsid and lysis characteristics differ from those of HPV, would not improve accuracy and is not recommended in current clinical guidelines.

4.⁠ ⁠“Impurities introduced during thawing/centrifugation may lower PCR sensitivity.”

Response

The thaw–vortex–centrifuge step removes mucous debris, thereby increasing PCR efficiency. Adequacy of every specimen was verified by the internal control signal; no inhibition was observed. Routine quality-control charts (cycle-threshold values of the internal control) remained within manufacturer specifications throughout the study period, supporting the purity and reliability of the extracted DNA.

If, after the corrections made according to the recommendations, the manuscript still needs extensive English proofreading, we will send it to the appropriate department within the publishing house.

We trust these clarifications address the reviewer’s concerns and thank them again for their constructive feedback.

Kind regards

Round 2

Reviewer 1 Report

Comments and Suggestions for Authors

The authors have satisfactorily addressed all major concerns raised in the initial review. The manuscript is now clearer, better organized, and the methods, results, and discussion sections are appropriately aligned. Minor language and style issues can be resolved during the journal’s editing stage.